# Anthocyanin Accumulation in Grape Berry Skin Promoted by Endophytic *Microbacterium* sp. che218 Isolated from Wine Grape Shoot Xylem

**DOI:** 10.3390/microorganisms12091906

**Published:** 2024-09-19

**Authors:** Yuka Teshigawara, Shiori Sato, Takayuki Asada, Masutoshi Nojiri, Shunji Suzuki, Yoshinao Aoki

**Affiliations:** 1Laboratory of Fruit Genetic Engineering, The Institute of Enology and Viticulture, University of Yamanashi, 1-13-1, Kofu 400-0005, Yamanashi, Japan; 2Agri-Bio Research Center, Kaneka Corporation, Iwata-shi 438-0802, Shizuoka, Japan; takayuki.asada@kaneka.co.jp (T.A.); masutoshi.nojiri@kaneka.co.jp (M.N.)

**Keywords:** actinobacterium, anthocyanin, endophyte, wine grape, *Microbacterium*, viticultural practice

## Abstract

Grape berry skin coloration is a key determinant of the commercial value of red wines. Global warming caused by climate change has inhibited anthocyanin biosynthesis in berry skins, leading to poor coloration. Through two-year field experiments, the endophyte che218 isolated from grape shoot xylem promoted anthocyanin accumulation in berry skins. The che218 enhanced anthocyanin biosynthesis in grapevine cultured cells. In the 2022 growing season, applying che218 to grape bunches enhanced anthocyanin accumulation in berry skins on day 20 post-treatment. However, the anthocyanin accumulation enhancing effect of che218 became negligible at harvest. In the 2023 growing season, che218 enhanced anthocyanin accumulation in berry skins on day 15 post-treatment and at harvest (day 30 post-treatment) and also upregulated the transcription of *mybA1* and *UFGT*, two genes that regulate anthocyanin biosynthesis in berry skins. Whole genome sequencing demonstrated that che218 is an unidentified *Microbacterium* species. However, it remains unknown how che218 is involved in the biosynthesis of anthocyanin in berry skins. This study provides insights into the development of an eco-friendly endophyte-mediated technique for improving grape berry skin coloration, thereby mitigating the effects of global warming on berry skin coloration.

## 1. Introduction

Grape berry skin coloration is a key determinant of the commercial value of red wines. Berry skin coloration is affected by anthocyanin content; berries with insufficient anthocyanin levels at harvest exhibit poor coloration, resulting in red wines having a less intense color [1]. Anthocyanins also play a role in stabilizing tannins, producing a pleasant mouthfeel in red wines [2]. Thus, anthocyanin content in berry skin significantly impacts the color and taste of red wines.

The global warming-induced rise in temperatures during the grape growing period has adversely affected anthocyanin biosynthesis in berry skins [3], leading to inadequate berry skin coloration. This has prompted the relocation of vineyards further northward or to areas with higher elevations [4]. In current wine grape cultivation regions, grape growers are confronted with the dilemma of whether to replace traditional red wine grape cultivars with cultivars that exhibit greater tolerance to high temperatures. Viticultural practices, such as trunk girdling [5], leaf removal [6], and cluster thinning [7], have been implemented, but these require long labor hours and technical skills.

An alternative approach to sustain berry skin coloration is the application of chemical stimulants, such as abscisic acid (ABA), on grape bunches [8]. ABA regulates anthocyanin biosynthesis in berry skins [9]. Treating grape bunches with ABA enhanced anthocyanin accumulation through the upregulation of a key enzyme in anthocyanin biosynthesis, UDP-glucose: flavonoid 3-*O*-glucosyltransferase, in berry skins [8]. The application of synthetic ABA (S-ABA) in field conditions has improved berry skin coloration [10]. In addition, S-ABA promoted anthocyanin biosynthesis through upregulating flavonoid synthesis gene expression in berry skins [11]. However, concerns over environmental pollution caused by the use of synthetic chemicals in viticulture cannot be dispelled. Considering environmental conservation in viticulture, it is necessary to develop techniques that use natural bioactive compounds to improve anthocyanin biosynthesis in berry skins. For example, vanillylacetone, a pungent compound found in ginger, enhanced anthocyanin biosynthesis in berry skins through the upregulation of the transcriptional levels of genes involved in anthocyanin biosynthetic pathways [12]. Thus, the application of eco-friendly natural stimulants that activate anthocyanin biosynthesis is an alternative strategy to maintain berry skin coloration under elevated temperature conditions.

The application of exogenous microorganisms induces anthocyanin biosynthesis in plants. For example, exposure to endophytic fungi promoted anthocyanin accumulation in berry skins by enhancing the phenylpropanoid biosynthesis pathway [13]. Treating *Arabidopsis thaliana* seedlings with volatile compounds released by *Trichoderma asperellum* IsmT5, which was isolated from the maize rhizosphere, induced threefold higher anthocyanin accumulation compared to the control (untreated seedlings) [14]. Furthermore, the endophytic bacterium *Azospirillum lipoferum* USA 59b produced ABA and gibberellins, which are involved in the alleviation of drought stress symptoms in maize [15]. Based on these findings, it was hypothesized that the application of endophytic microbes in wine grapes would promote anthocyanin biosynthesis in berry skins. Aiming to explore anthocyanin biosynthesis-promoting endophytic bacteria isolated from the shoot xylem of wine grapes, this study demonstrated that *Microbacterium* sp. che218 promoted anthocyanin accumulation in berry skins of field-grown grapevines in the 2022 and 2023 growing seasons.

## 2. Materials and Methods

### 2.1. Plant Materials

Grapevine cultured cells, VR cells, prepared from of *Vitis* interspecific hybrid cultivar Bailey Alicante A [16], were obtained from the RIKEN BioResource Center (Tsukuba, Ibaraki, Japan) and used for screening anthocyanin biosynthesis-promoting endophytic bacteria. The cultured cells were maintained on a modified Linsmaier and Skoog (LS) agar medium (pH 5.7) [17] supplemented with 3% sucrose, 0.2 mg/mL kinetin, 0.05 mg/mL 2,4-dichlorophenoxyacetic acid and 0.5% gellan gum at 27 °C in the dark and subcultured every 7 d.

Muscat Bailey A [*Vitis labruscana* (Bailey) × *Vitis vinifera* (Muscat Hamburg)] was cultivated in the experimental vineyard of The Institute of Enology and Viticulture, University of Yamanashi, Japan (latitude 35.6800524, longitude 138.569268, elevation 250 m). The grapevines were approximately 30 years old, grafted onto rootstock 5BB and trained in the Guyot-style. Weather data for Kofu, where the experimental vineyard was located, were obtained from the website of the Japan Meteorological Agency (https://www.data.jma.go.jp/stats/etrn/index.php, accessed on 24 July 2024). Average temperatures, maximum and minimum temperatures, growing degree days (GDD, base threshold of 10 °C) and precipitation from 1 April through 31 October were summarized monthly for the 2022 and 2023 growing seasons (Appendix A).

### 2.2. Identification of Anthocyanin Biosynthesis-Promoting Endophytic Bacteria from Grapevine Shoot Xylems

Endophytic bacteria were isolated from grapevine shoot xylems as described previously [18]. Briefly, shoots of grapevines (*Vitis* sp. cv. Koshu, *V. vinifera* cvs. Pinot Noir, Chardonnay, and Cabernet Sauvignon) were surface sterilized with sodium hypochlorite solution, and then bark and epidermal tissues were peeled off from the shoots using a sterilized knife. Xylem tissues were shaved using a sterilized grater and shaken in phosphate buffer (pH 7.4). The filtrate was incubated in a soybean casein digest (SCD) agar plate at 25 °C for 3 d after removing xylem debris by filtration. Finally, 60 bacterial endophytes were isolated on the plates. Each endophytic bacterium was incubated in an SCD medium for 1 to 3 d at 37 °C. The cultures were prepared at 1.0 × 10^8^ CFU/mL with sterile water. VR cells were incubated on the modified LS agar medium in 24-well plates at 27 °C in the dark for 5 d. As a control, ABA (Tokyo Chemical Industry Co., Ltd., Tokyo, Japan) and culture supernatant of che218, isolated as an anthocyanin biosynthesis-promoting endophyte, were used. The supernatant of the 1-day culture of che218 was collected by centrifugation at 30,000 rpm for 3 min and further refined by filter sterilization using a Minisart single-use filter (0.2 μm pore, Sartorius, Göttingen, Germany). Twenty μL of each endophytic bacterial culture, che218 culture, 1 mM ABA, or the supernatant of che218 was dropped on a VR cell mass. To avoid anthocyanin accumulation by drying, the 24-well plates were placed in a plastic box containing a moistened paper towel and incubated at 27 °C for 5 d under light irradiation (96.2 µmol·m^−2^·s^−1^/16 h/d). The cells were subjected to anthocyanin measurement. Finally, the best candidate promoting anthocyanin accumulation in VR cells, endophyte che218, was selected and subjected to field experiments.

### 2.3. Field Experiments 

Bunches of two field-grown grapevines were treated with che218 in the 2022 and 2023 growing seasons. Briefly, che218 was incubated in SCD medium for 1 d at 37 °C. The cultures were prepared at 1.0 × 10^8^ CFU/mL with sterile water. The culture of endophytic bacterium koe227, which did not promote anthocyanin accumulation in VR cells, was selected as the control and prepared by the same method as that for che218 preparation. The supernatant of the 1-day culture of che218 was prepared as mentioned above. The supernatant was diluted five times with SCD medium. A fivefold diluted SCD medium was used as the control. Approach BI (Kao, Wakayama, Japan) was added as a spreader to all treatment solutions to make a final concentration of 0.1%. The treatment was carried out at véraison (at 7 a.m. on 23 August 2022 and 4 August 2023, respectively), the treatment was carried out by placing 200 mL of each treatment solution in a plastic cup and immersing each grape bunch on two grapevines in it. The weather was fine before and after the treatment. Three bunches were collected for each treatment on days 0, 20 and 30 post-treatment in the 2022 growing season or days 0, 15 and 30 post-treatment in the 2023 growing season.

### 2.4. Anthocyanin Measurement

Total anthocyanin concentration in che218-treated VR cells and berry skins was measured following the described methods [19,20].

VR cells treated with che218 for 5 d were collected in 2 mL microtubes and weighed. Seven hundred μL of 1% HCl-methanol was added into the microtubes. After an overnight incubation at room temperature in the dark, the supernatants were obtained by centrifugation at 6000 rpm for 5 min. Absorbance (OD_520_) of the supernatants was measured using a spectrometer (UV-1800, Shimadzu, Kyoto, Japan).

Twenty berries (six from the top of the bunch, eight from the middle of the bunch, and six from the bottom of the bunch) from each bunch were collected at the indicated period after the treatment with che218. Skins were peeled off from the berries by tweezers and placed in a mortar containing liquid nitrogen. Pulverized skin samples were prepared by homogenizing the skins with a pestle. One gram of pulverized sample was macerated in a 15 mL tube containing 10 mL of 1% HCl-methanol and incubated at room temperature overnight in the dark. After centrifugation at 3000 rpm for 3 min, the supernatant was filtered through a 0.45 μm cellulose acetate filter (Advantec Toyo, Tokyo, Japan). The absorbance (OD_520_) of the filtrate was measured using the spectrometer.

Total anthocyanin concentration was calculated from the OD_520_ value by a previously published formula [19] and converted into malvidin-3-glucoside equivalent as μg per gram of fresh weight of VR cells or mg per gram of fresh weight of berry skins, respectively.

### 2.5. Total RNA Isolation

Total RNA was extracted from skins collected from berries on days 0, 15 or 30 post-treatment in the 2023 growing season. Briefly, skins peeled off from the berries by tweezers were frozen in liquid nitrogen and homogenized with a pestle. Total RNA isolation from the pulverized skin samples was performed with a Fruit-mate for RNA Purification (Takara, Otsu, Japan) and an RNeasy Plant Mini Kit (Qiagen, Hilden, Germany) using an automated QIAcube (Qiagen) following the manufacturer’s instructions.

### 2.6. cDNA Synthesis

First-strand cDNA was synthesized from total RNA using a PrimeScript RT Reagent Kit with gDNA Eraser (Takara) following the manufacturer’s instructions.

### 2.7. Quantitative RT-PCR

Quantitative RT-PCR was performed with TB Green Premix EX Taq II (Tli RNaseH Plus) (Takara) with a Thermal Cycler Dice Real-Time System TP980 (Takara) following the manufacturer’s instructions. Briefly, PCR amplification was performed for 40 cycles at 95 °C for 5 s and at 60 °C for 45 s, and for 1 cycle at 95 °C for 15 s, at 60 °C for 30 s, and at 95 °C for 15 s after an initial denaturation at 95 °C for 30 s. The nucleotide sequences of the primers used for quantitative RT-PCR were as follows: Myb-related transcription factor (MybA1) primers (5′-GCAAGCCTCAGGACAGAAGAA-3′ and 5′-ATCCCAGAAGCCCACATCAA-3′ from *VvMybA1*, accession no. AB111101), UDP glucose flavonoid 3-O-glucosyl transferase (UFGT) primers (5′-CTTCTTCAGCACCAGCCAATC-3′ and 5′-AGGCACACCGTCGGAGATAT-3′ from *V. vinifera* UFGT, accession no. NM_001397857), 9-cis-epoxycarotenoid dioxygenase 1 (NCED1) primers (5’-GAGACCCCAACTCTGGCAGG-3′ and 5′-AAGGTGCCGTGGAATCCATAG-3′ from *V. vinifera* NCED1, accession no. NM_001281270) and β-actin primers (5′-CAAGAGCTGGAAACTGCAAAGA-3′ and 5′-AATGAGAGATGGCTGGAAGAGG-3′ from *V. vinifera* actin 1, accession no. AF369524). The dissociation curves were evaluated to verify the specificity of the amplification reaction. Using the standard curve method and Thermal Cycler Dice Real Time System Software (ver. 5.11, Takara), the expression level of each gene was determined as the number of amplification cycles needed to reach a fixed threshold. Data are expressed as relative values to β-actin and presented as means ± standard errors.

### 2.8. Identification of che218 by Whole Genome Sequencing

Genomic DNA was extracted from the 3-day culture of che218 in the SCD medium using a Genomic-tip 100G (Qiagen, Hilden, Germany) following the manufacturer’s instructions. Sequencing using Revio (Pacific Biosciences, Menlo Park, CA, USA) and assembling using Flye (ver. 2.9.2-b1786) were performed by Bioengineering Lab. Co., Ltd. (Kanagawa, Japan), as detailed in the previous paper [21].

The 16S rDNA sequences of 50 samples closest to che218 16S rDNA were collected from the NCBI database. *Bacillus subtilis* subsp. subtilis str. 168 (accession no. CP053102) was used as an outgroup. Phylogenetic analysis of the 16S rDNA sequences was performed by the maximum likelihood method using Molecular Evolutionary Genetics Analysis software, MEGA11 (available from https://www.megasoftware.net/, accessed on 7 June 2024). Average Nucleotide Identity (ANI) analysis of che218 whole genome sequence with *Microbacterium enclense* bin7 and *Microbacterium binotii* Au-Mic3 whole genomes (accession no. CP116226 and CP090347, respectively) was performed by ANI Calculator (available from https://www.ezbiocloud.net/tools/ani, accessed on 7 June 2024). Whole genome sequences of che218, *M. enclense* bin7 and *M. binotii* Au-Mic3 were also subjected to comparative genome analysis using DiGAlign [22].

### 2.9. Statistical Analysis

Data are presented as means ± standard errors of biological replicates. Statistical analysis was performed by Dunnett’s test using Excel statistics software 2012.

## 3. Results

### 3.1. Effect of che218 Treatment on Anthocyanin Accumulation in Grapevine Cultured Cells

Screening for anthocyanin biosynthesis-promoting bacteria from among 60 endophytic bacteria isolated from shoot xylems of wine grapes yielded che218. che218 was isolated from Chardonnay. Similar to ABA treatment, che218 treatment enhanced anthocyanin accumulation in VR cells compared with control cells and SCD-treated cells (Figure 1A,B). VR cells treated with the supernatant of the 1-day culture of che218 also accumulated more anthocyanin than control cells and SCD-treated cells. The effect of che218 on anthocyanin accumulation in VR cells was dose dependent. che218 at a cell density of 1.0 × 10^8^ CFU/mL but not 1.0 × 10^6^ or 1.0 × 10^7^ CFU/mL increased anthocyanin content in VR cells compared with SCD-treated cells (Figure 1C). 

Taken together, the results suggest that che218 promotes anthocyanin biosynthesis in VR cells. 

### 3.2. Effect of che218 Treatment on Anthocyanin Accumulation in Berry Skins of Field-Grown Grapevines

Bunches of field-grown grapevines were treated with che218 (1.0 × 10^8^ CFU/mL) at véraison in the 2022 and 2023 growing seasons. Berry skin coloration enhancement by che218 was visible to the naked eye on day 20 post-treatment in the 2022 growing season (Figure 2A). The che218-treated bunches on day 20 post-treatment accumulated more anthocyanin in berry skins than the SCD-treated bunches (Figure 2B). The effects of che218 on anthocyanin accumulation in berry skins became negligible on day 30 post-treatment.

In the 2023 growing season, che218 also enhanced berry skin coloration (Figure 3). Enhanced berry skin coloration was also visible to the naked eye on days 15 and 30 post-treatment (Figure 3A). The che218-treated bunches accumulated more anthocyanin in berry skins than the SCD-treated bunches on days 15 and 30 post-treatment (Figure 3B). The supernatant of the 1-day culture of che218 and 1.0 × 10^8^ CFU/mL of koe227 did not affect anthocyanin accumulation in berry skins (Figure 3B). 

The field used in this study belonged to Region V on the Winkler Index (Appendix A). GDD in the 2022 and 2023 growing seasons demonstrated that the grapevines tested were cultivated under extremely high temperatures from July to September. A comparison of temperatures for the critical grape ripening period from July to September revealed higher temperatures in the 2023 growing season compared to the 2022 growing season. The temperature difference during the key phenological stages of véraison and berry ripening may have significant implications for anthocyanin accumulation and overall berry development. The effect of che218 on berry skin coloration at harvest (on day 30 post-treatment) varied from year to year (Figure 2B and Figure 3B). Analysis of temperature data suggests that the higher temperatures observed in the 2023 growing season likely resulted in reduced anthocyanin accumulation at harvest. However, treatment with che218 appeared to mitigate this negative effect, effectively restoring anthocyanin levels. This observation implies that che218 may play a protective role against temperature-induced reductions in anthocyanin synthesis, potentially offering a valuable strategy for maintaining grape quality under varying climatic conditions.

### 3.3. Effect of che218 Treatment on the Transcription of Anthocyanin Biosynthesis-Related Genes in Berry Skins of Field-Grown Grapevines

Anthocyanin biosynthesis in berry skins is regulated by the expression of *mybA1*, which encodes a transcription factor that controls the expression of *UFGT*, a gene that encodes an enzyme catalyzing the glycosylation of anthocyanidins [23]. To determine whether che218 upregulated the transcription of *mybA1* and *UFGT*, the expression levels of *mybA1* and *UFGT* in berry skins treated with che218 were investigated in the 2023 growing seasons. The transcripts of *mybA1* and *UFGT* in che218-treated berry skins were more abundant than those of SCD-treated bunches on days 15 and 30 post-treatment (Figure 4).

These results suggest that the acceleration of anthocyanin accumulation in berry skins by che218 occurs through the upregulation of *mybA1* transcription.

### 3.4. che218 Is a Microbacterium sp.

To shed light on the mechanisms underlying the promotion of anthocyanin biosynthesis by che218, genome sequencing of che218 was conducted [21]. The circular genome consists of a single contig of 3,591,470 bp (accession no. AP031501). The GC content of 70.4% is a distinctive feature of the che218 whole genome sequence. Annotation by DFAST (ver. 1.2.0) [24] predicted 3387 coding sequences, 9 rRNA genes and 52 tRNA genes. The che218 genome has no anthocyanin biosynthesis-related genes in plants. For example, genes encoding enzymes for ABA biosynthesis, such as sesquiterpene synthases, were not detected in the che218 genome, as reported in filamentous fungi [25].

Homological analysis of the 16S rDNA nucleotide sequence of che218 using NCBI BLAST showed 100% identity with a corresponding sequence of *M. enclense* bin7 and high homologies with the corresponding sequences of the *Microbacterium* species. Phylogenetic analysis of the 16S rDNA nucleotide sequence of che218 and *Microbacterium* species revealed that che218 formed a cluster with *M. enclense* bin7 (Figure 5).

Average nucleotide identity (ANI) is a key computational analysis tool for defining species boundaries of bacteria [26]. ANI analysis of che218 whole genome sequence and the whole genome sequences of *M. enclense* bin7 in the same cluster on the phylogenetic tree and *M. binotii* Au-Mic3 in the neighboring cluster on the phylogenetic tree (Figure 5) gave OrthoANIu values of 82.97% and 76.18%, respectively. Because the proposed cutoff for species boundaries is 95 to 96% [27], che218 was determined to be an unknown species. Comparison of che218 whole genome sequence with the whole genome sequences of *M. enclense* bin7 and *M. binotii* Au-Mic3 using DiGAlign revealed significant differences among the whole genome structures and sequences even though the 16S gene is 100% identical (Figure 6).

Taken together, the results suggest that che218 is an unidentified *Microbacterium* sp.

## 4. Discussion

Anthocyanin accumulation in che218-treated berry skins was dependent on *mybA1* expression and the subsequent *UFGT* expression (Figure 4). Exogenous ABA application markedly affected anthocyanin biosynthesis by upregulating *mybA1* and *UFGT* expression [27]. The increase of endogenous ABA content in berry skin triggered berry skin coloration [28]. Whole genome sequencing of che218 demonstrated that che218 cannot produce ABA per se, whereas some bacteria can produce ABA themselves. For example, *Azospirillum brasilense* sp. 245, a plant-growth-promoting bacterium, produces ABA in a culture medium [29]. Xanthoxin, which is produced by *Bacillus marisflavi* and is an intermediate in the biosynthesis of ABA, induced drought stress tolerance in plants similar to ABA [30]. The che218 culture supernatant enhanced anthocyanin accumulation in grapevine-cultured cells (Figure 1B). However, this effect could not be confirmed in the field experiment performed in the 2023 growing season. Although this discrepancy in results may be related to the quality and quantity of ABA analogous compounds produced by che218, it is not reasonable to conclude that che218 directly secretes the compounds. Anthocyanin accumulation in berry skin was also enhanced by exogenous ethylene application through the upregulation of *mybA1* and *UFGT* expression [31]. Future studies employing integrated genomic (che218)-metabolomic (che218 culture supernatant and/or che218-treated berry skins) analysis are expected to reveal how che218 upregulates *mybA1* and *UFGT* expression. It was reported that the application of exogenous stimulants to berry skin activates ABA biosynthesis [12]. Further investigation on the temporal profiles of ABA and related plant hormones in che218-treated berry skins may yield new information on the mechanisms underlying the microbial enhancement of anthocyanin biosynthesis in plants.

Whole genome sequencing of che218 suggested that it is an unidentified *Microbacterium* species. The genus *Microbacterium* belongs to a high GC actinobacterial taxon and comprises more than 90 species that have been isolated from a wide range of habitats and hosts [32]. Endophytic *Microbacterium* is often isolated from plants as a novel species. For example, *M. zeae* sp. nov. was isolated as an endophytic actinobacterium from surface-sterilized stem tissue of maize planted in China [33]. *M. endophyticum* sp. nov. and *M. halimionae* sp. nov. were isolated from the salt marsh plant and defined as a novel genus [34]. A comprehensive analysis of *Microbacterium* strains and their most prominent plant growth-promoting traits showed that many *Microbacterium* strains produced siderophores, 1-aminocyclopropane-1-carboxylic acid deaminase and solubilized phosphate [32]. A total of 17 of the 29 strains tested produced indole-3-acetic acid (IAA), and 16 of the 17 strains had both genes for the synthesis of tryptophan, the main precursor of IAA, and genes necessary for the conversion of tryptophan into IAA through the tryptamine pathway. Interestingly, che218 has IAA biosynthesis-related genes. Whole genome sequencing demonstrated that che218 had genes encoding anthranilate phosphoribosyltransferase (EC 2.4.2.18), tryptophan synthase alpha chain (EC 4.2.1.20), and tryptophan synthase beta chain (EC 4.2.1.20) for tryptophan biosynthesis, and genes encoding tryptophan decarboxylase (EC 4.1.1.28) and flavin-containing monoamine oxidase (EC 1.4.3.4) for IAA biosynthesis through the tryptamine pathway. The application of exogenous IAA-producing *Microbacterium* to plants promotes plant growth. For example, *M. albopurpureum* ET2^T^ was isolated from the rhizoplane of *Chiloschista parishii* Seidenf. produced IAA that promoted the rooting of kidney beans and shoot growth of garden cress and wheat [35]. Unfortunately, this study could not present results related to the plant growth-promoting activity of che218. Future studies on plant growth promotion and IAA production by che218 would reveal new roles of che218 other than promoting anthocyanin accumulation in berry skins.

The increase in average temperatures over the past several decades in wine grape-producing regions around the world has resulted in the reduction of anthocyanin content in berry skin [36]. Trunk girdling [5], leaf removal [6], cluster thinning [7] and ABA application [8] have been adopted to improve berry skin coloration. However, no endophyte-mediated techniques for promoting berry skin coloration have been developed. The potential application of endophytes for improving wine grape berry coloration presents both opportunities and challenges that require careful consideration. First, while endophytes are naturally occurring entities, their introduction and accumulation in an ecosystem can modify the environment in ways that are not fully understood. Therefore, their environmental impact should be thoroughly assessed before claiming any environmental benefits. Second, endophytes have the potential to establish themselves in the environment, which could lead to long-term effects. The colonization of plants by biological control agents, including endophytes, may contribute to plant protection against diseases. For example, the suppressive activity of endophytes against phytopathogenic fungi is influenced by their capacity to colonize in plants [37]. It is reasonable to assume that endophytes show a greater affinity for plants than other environmental microorganisms. However, so far, this study cannot demonstrate any positive results of che218 colonization on bunches. Field trials of che218 application to bunches would reveal whether che218 can adapt, proliferate and exert its effects over a relatively long period on bunches. On the other hand, it is crucial to note that the distinction between beneficial and potentially harmful microorganisms can be subtle and subject to change under different conditions. Consequently, extensive research and rigorous testing are necessary to ensure the safe and effective use of che218 in viticulture. Endophytes invade plants from the soil, primarily through the roots, and then colonize the internal plant parts [38]. Future investigations on the uptake of che218 from root into grapevine seedlings or the physical injection of che218 into grapevine seedlings are also expected to provide information on the production of high-value-added grapevine seedlings using endophytes.

In conclusion, this study has demonstrated the significant potential of the endophytic bacterium che218 in enhancing anthocyanin accumulation in grape berries. This study focused primarily on the effects of che218 on anthocyanin biosynthesis. However, the influence of che218 on overall grape berry quality and other secondary metabolites merits additional investigation. Future research directions should include a more comprehensive analysis of the phenylpropanoid pathway, examining the expression of various genes involved in this pathway. For example, ABA has been shown to promote anthocyanin biosynthesis through the upregulation of flavonoid synthesis gene expression in berry skins [11]. This broader approach would provide a more complete understanding of che218’s effects on grape berry qualities. Additionally, investigating the impact of che218 on other important compounds in grape berries, such as resveratrol and its derivatives, could reveal further benefits of che218. These findings underscore the complex interplay between plant hormones, endophytic bacteria, and secondary metabolism in grape berries, highlighting the need for continued research in this area. The potential applications of che218 in viticulture are promising, offering a natural means to enhance grape and wine qualities. This work not only contributes to the understanding of plant-microbe interactions, but also opens new avenues for sustainable practices in agriculture and food production.

## Figures and Tables

**Figure 1 microorganisms-12-01906-f001:**
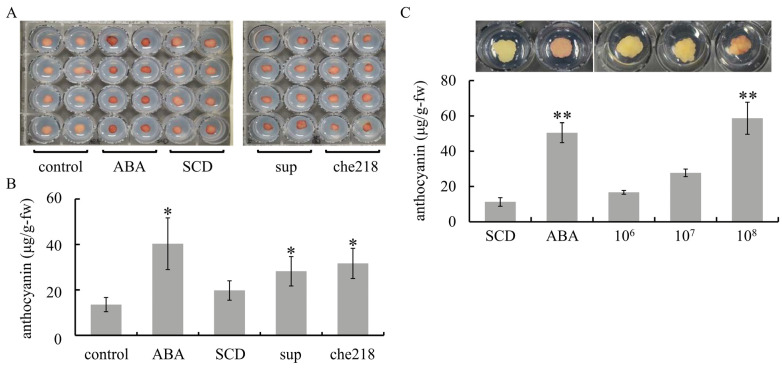
Effect of che218 on anthocyanin accumulation in grapevine cultured cells. (**A**) VR cells were treated with 1.0 × 10^8^ CFU/mL of che218. (**B**) Anthocyanin content in VR cells treated with 1.0 × 10^8^ CFU/mL of che218 on day 5 post-treatment. Bars indicate means ± standard errors for eight VR cells. (**C**) Dose dependence of anthocyanin biosynthesis-promoting activity of che218. Representative VR cells are shown above the graph. Bars indicate means ± standard errors for eight VR cells. * *p* < 0.05 compared with the control (Dunnett’s test). ** *p* < 0.01 compared with the control (Dunnett’s test). control, no treatment. ABA, treated with 1 mM ABA. SCD, treated with a fivefold diluted SCD medium. sup, treated with supernatant from 1-day culture of che218. che218, treated with 1.0 × 10^8^ CFU/mL of che218 culture. 10^6^, treated with 1.0 × 10^6^ CFU/mL of che218. 10^7^, treated with 1.0 × 10^7^ CFU/mL of che218. 10^8^, treated with 1.0 × 10^8^ CFU/mL of che218.

**Figure 2 microorganisms-12-01906-f002:**
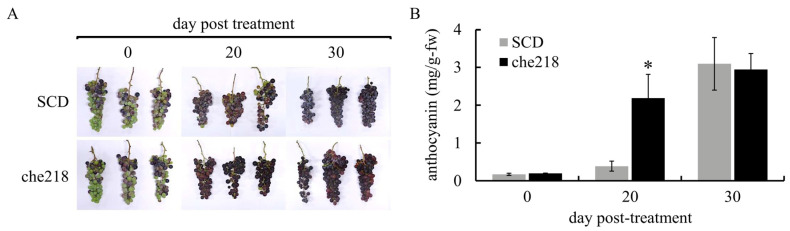
Effect of che218 on anthocyanin accumulation in grape berry skins in the 2022 growing season. Field-grown grape bunches were treated with 1.0 × 10^8^ CFU/mL of che218 or fivefold diluted SCD medium (control). (**A**) Photographs of bunches. (**B**) Anthocyanin contents in berry skins. Three bunches were collected at the indicated times post-treatment and subjected to anthocyanin measurement. Data indicate means ± standard errors for three bunches. * *p* < 0.05 compared with SCD-treated bunches (Dunnett’s test). SCD, treated with a fivefold diluted SCD medium. che218, treated with 1.0 × 10^8^ CFU/mL of che218.

**Figure 3 microorganisms-12-01906-f003:**
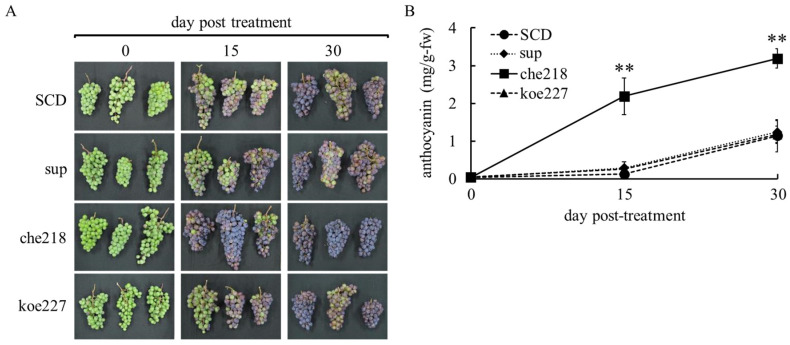
Effect of che218 on anthocyanin accumulation in grape berry skins in the 2023 growing season. Field-grown grape bunches were treated with 1.0 × 10^8^ CFU/mL of che218, fivefold diluted SCD medium (control), supernatant from 1-day culture of che218 or 1.0 × 10^8^ CFU/mL of koe227. (**A**) Photographs of bunches. (**B**) Anthocyanin contents in berry skins. Three bunches were collected at the indicated days post-treatment and subjected to anthocyanin measurement. Data indicate means ± standard errors for three bunches. ** *p* < 0.01 compared with SCD-treated bunches (Dunnett’s test). SCD, treated with a fivefold diluted SCD medium as control. sup, treated with supernatant from 1-day culture of che218. che218, treated with 1.0 × 10^8^ CFU/mL of che218. koe227, treated with 1.0 × 10^8^ CFU/mL of koe227, an endophytic bacterium that did not promote anthocyanin biosynthesis in VR cells.

**Figure 4 microorganisms-12-01906-f004:**
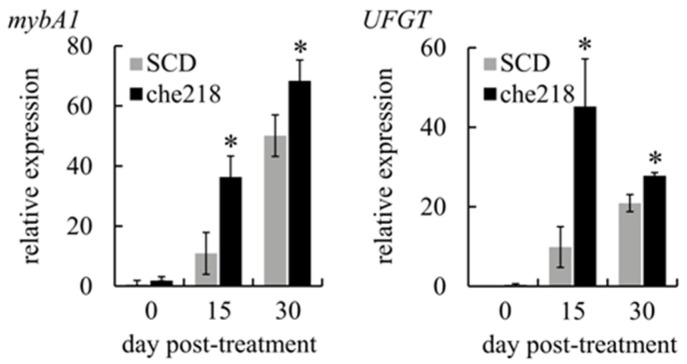
Effect of che218 on *mybA1* and *UFGT* transcription in grape berry skins. Field-grown grape bunches treated with 1.0 × 10^8^ CFU/mL of che218 or fivefold diluted SCD medium (control) in the 2023 growing season were subjected to quantitative RT-PCR. Data were calculated as gene expression relative to β-actin expression. Bars indicate means ± standard errors for three bunches. * *p* < 0.05 compared with SCD-treated bunches (Dunnett’s test). SCD, treated with a fivefold diluted SCD medium. che218, treated with 1.0 × 10^8^ CFU/mL of che218.

**Figure 5 microorganisms-12-01906-f005:**
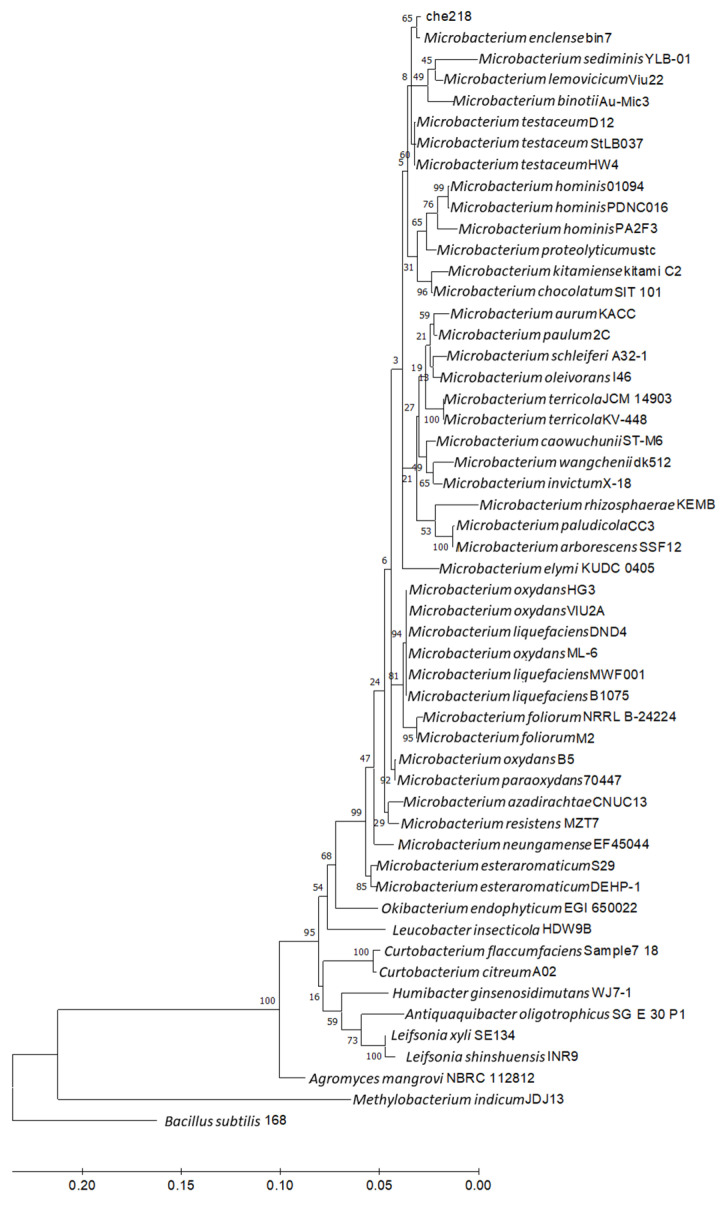
Phylogenetic analysis of 16S rDNA nucleotide sequence of che218. Information on *Microbacterium* species and other bacteria used in the phylogenetic analysis is listed in Appendix A. *Bacillus subtilis* subsp. *subtilis* str. 168 (CP053102) was used as an outgroup. A phylogenetic tree was designed by means of the maximum likelihood method using MEGA 11. Bar indicates 1% band dissimilarity. The distance corresponds to the number of nucleotide substitutions per site.

**Figure 6 microorganisms-12-01906-f006:**
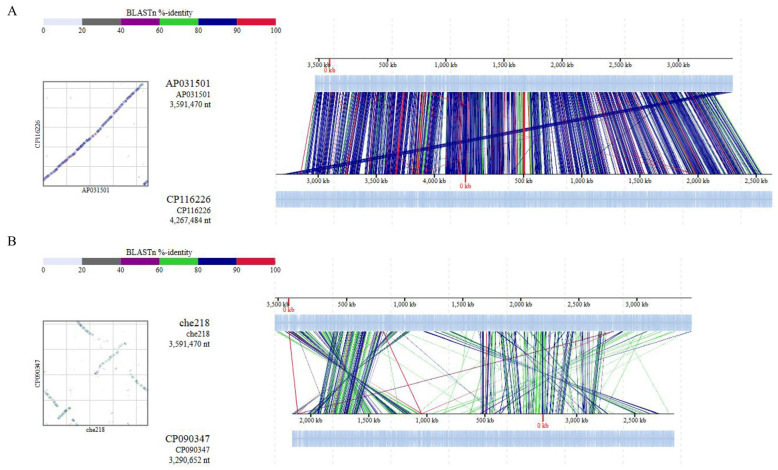
Comparison of whole genome sequences by ANI analysis. Whole genome sequence alignments of che218, *M. enclense* isolate bin7 (**A**) and *M. binotii* strain Au-Mic3 (**B**) were visualized with a genome alignment viewer, DiGAlign [22]. High-similarity regions detected by BLASTn were color-coded on the basis of the indicated percentage identity. Dot plots summarizing these high-similarity regions are shown beside the alignment.

## Data Availability

The Whole Genome of che218 has been deposited in DDBJ/ENA/GenBank under the accession numbers AP031501 (whole genome), PRJDB17835 (BioProject), SAMD00769561 (BioSample), DRX530437 (Experiment) and DRR546667 (raw sequencing reads).

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
