# Peer review of "Anthocyanin Accumulation in Grape Berry Skin Promoted by Endophytic Microbacterium sp. che218 Isolated from Wine Grape Shoot Xylem"

_microorganisms, 2024, doi:10.3390/microorganisms12091906_

Round 1

Reviewer 1 Report

Comments and Suggestions for Authors

The paper is within the scope of the journal, the methods used are appropriate and reproducible (I have added comments about some methods that should be briefly detailed in the document instead of just adding a reference - see comment below.), and the results and discussion will be of great interest to the readership. Figures are adequate and appropriate, well presented and clear. The paper is interesting and presents a rich collection of information on the use of endophyte-mediated techniques for promoting berry skin coloration in grapes. The authors investigated the ability of endophytes to increase anthocyanin accumulation in cell cultures in the laboratory and in berry skins of field-grown grapevines. In addition, they also suggest future experiments to follow the use of endophytic microbes in wine grapes to improve berry quality. In some parts of the document the authors mention the impact of an increase in temperatures over the past few decades in wine grape-producing regions around the world and that one of the impacts would be the reduction of anthocyanin content in berry skin - so it is necessary to have the detailed weather conditions over the seasons the field experiment was performed - this is an important information that was missing in the document. I support the publication of this document after the required revisions. Below are some points that I have suggested to the authors for inclusion/clarification in the document:

L 45: …on grape bunches (Please add a reference to support this sentence).

L 50: berry skin coloration [10] and increased CIRG index, polyphenols, anthocyanins and sugars in Cabernet Sauvignon grapes [https://doi.org/10.1590/S0100-29452012000200003].

L 78: … Linsmaier and Skoog (Please add the medium reference and the amount of agar used)

L 78: How about subculture interval?

L 85: How about the rootstock? Also please add the climate conditions of the region

L 88-89: “Endophytic bacteria were isolated from grapevine shoot xylems as described previously” – I know you added a reference for this, but please also briefly describe the protocol.

L 94: “che218” – This is the first time you mentioned it in the materials and methods - so please give more details about it.

L 96: or the d che218 culture was dropped on a VR cell mass.

L 104-115: Double check the text format

L 104: Muscat Bailey A? Add the cultivar name

L 105: How about the amount of inoculum?

L 112: What time of day was the treatment performed? What were the weather conditions before and after the treatment?

L 115: part of the text is missing - please check it again.

L 124: Was there a specific berry position in the bunch for sampling?

L 133: Consider adding a formula instead of just mentioning it.

It is important to add the climatic conditions during the seasons in which the field experiment was conducted.

L 190: Consider adding the data in the supplementary material

L 213: Same comment as L 104

L 214: No need to add a new paragraph

L 320: How about the weather conditions this season? Was it different from the previous season?

L 361: to the best of our knowledge,

Add a final sentence highlighting the importance of the findings over these years of this beautiful research...

Author Response

Review  #1

Comments and Suggestions for Authors

The paper is within the scope of the journal, the methods used are appropriate and reproducible (I have added comments about some methods that should be briefly detailed in the document instead of just adding a reference - see comment below.), and the results and discussion will be of great interest to the readership. Figures are adequate and appropriate, well presented and clear. The paper is interesting and presents a rich collection of information on the use of endophyte-mediated techniques for promoting berry skin coloration in grapes. The authors investigated the ability of endophytes to increase anthocyanin accumulation in cell cultures in the laboratory and in berry skins of field-grown grapevines. In addition, they also suggest future experiments to follow the use of endophytic microbes in wine grapes to improve berry quality. In some parts of the document the authors mention the impact of an increase in temperatures over the past few decades in wine grape-producing regions around the world and that one of the impacts would be the reduction of anthocyanin content in berry skin - so it is necessary to have the detailed weather conditions over the seasons the field experiment was performed - this is an important information that was missing in the document. I support the publication of this document after the required revisions. Below are some points that I have suggested to the authors for inclusion/clarification in the document:

Answer: Thank you very much for your review to improve our manuscript. The following is our point-by-point response to the comments and detailing all changes made on the revised manuscript.

L 45: …on grape bunches (Please add a reference to support this sentence).

Answer: We added a reference ‘8. Peppi, M.C.; Walker, M.A.; Fidelibus, M.W. Application of abscisic acid rapidly upregulated UFGT gene expression and improved color of grape berries. Vitis 2008, 47, 11-24.’for the sentence.

                (See p. 2, line 45 in the revised manuscript, please)

L 50: berry skin coloration [10] and increased CIRG index, polyphenols, anthocyanins and sugars in Cabernet Sauvignon grapes [https://doi.org/10.1590/S0100-29452012000200003].

Answer: Thank you for your suggestion. Since this sentence explains synthetic ABA (commercial name: ProTone), we determined that it would not be appropriate to add a paper that used general ABA as references.

L 78: … Linsmaier and Skoog (Please add the medium reference and the amount of agar used)

Answer: Thank you for your suggestion. We added 0.5% gellan gum for LS agar medium. The information was added in the revised manuscript.

               (See p. 2, line 82 in the revised manuscript, please)

                Also,  a reference (Reuther 1982) was added for modified LS medium.

                (See the revised reference number [17] , please )

                Reuther, G.; (1982) Gewebekultur. Die Vermehrung von Pelargonienmutterpflanzen. Gb+Gw 32, 727-734.

L 78: How about subculture interval?

Answer: We subcultured the cells every 7 days. We added the information in the revised manuscript.

                (See p. 2, line 82 in the revised manuscript, please)

L 85: How about the rootstock? Also please add the climate conditions of the region

Answer: The grapevines were grafted onto rootstock 5BB. The information was added in the revised manuscript.

                (See p. 2, line 86 in the revised manuscript, please)

               Average temperatures, maximum and minimum temperatures, GDD, and precipitation from April 1 through October 31 in the experimental field in 2022 and 2023 growing seasons were added as supplemental Table S1 in the revised manuscript. We mentioned how the temperature differences between the two years affected anthocyanin accumulation promoted by che218.

                (See p. 2, lines 87-92 and p. 6, lines 243-257  in the revised manuscript and supplementary Table S1, please)

L 88-89: “Endophytic bacteria were isolated from grapevine shoot xylems as described previously” – I know you added a reference for this, but please also briefly describe the protocol.

Answer: Thank you for your suggestion. We added ‘brief information’ related to isolation of endophytic bacteria in the revised manuscript as follows:

                ‘Briefly, shoots of grapevines (Vitis sp. cv. Koshu, V. vinifera cvs. Pinot Noir, Chardonnay, and Cabernet Sauvignon) were surface-sterilized with sodium hypochlorite solution and then bark and epidermal tissues were peeled off from the shoots using a sterilized knife. Xylem tissues were shaved using a sterilized grater and shaken in phosphate buffer (pH 7.4). The filtrate was incubated in soybean casein digest (SCD) agar plate at 25 ºC for 3 d after removing xylem debris by filtration. Finally, 60 bacterial endophytes were isolated on the plates.’

                (See p. 3, lines 96-102 in the revised manuscript, please)

L 94: “che218” – This is the first time you mentioned it in the materials and methods - so please give more details about it.

Answer: We added the brief detail of che218 in the revised manuscript as follows:

‘culture supernatant of che218, isolated as an anthocyanin biosynthesis-promoting endophyte as mentioned later, were used.’

                (See p. 3, lines 106-107 in the revised manuscript, please)

L 96: or the d che218 culture was dropped on a VR cell mass.

Answer: We changed the sentence as follows:

‘Twenty μL of each endophytic bacterial culture, che218 culture, 1 mM ABA, or the supernatant of che218 was dropped on a VR cell mass’.

                (See p. 3, lines 110-112 in the revised manuscript, please)

L 104-115: Double check the text format

Answer: Thank you for your feedback. The italics were a mistake, so we have corrected them.

                (See the revised Section 2.3, please)

L 104: Muscat Bailey A? Add the cultivar name

Answer: Muscat Bailey A  is  a cultivar name. Muscat Bailey A is a hybrid grape cultivar [Vitis labrusca (Bailey) × Vitis vinifera (Muscat Hamburg)]. The cross-hybridization was developed in Japan in 1927 and, at present, Muscat Bailey A is Japan's most widely produced red wine grape cultivar.

                The scientific name of Muscat Bailey A  was listed in the revised manuscript.

                (See p. 2, line 83 in the revised manuscript, please)

L 105: How about the amount of inoculum?

Answer: The concentration of che218 inoculum solution were a cell density of 1.0 × 108 CFU/mL. Since all grape bunches on two grapevines were dipped into che218 inoculum solution, the exact amount of che218 on each bunch was unknown.  The information related to inoculum methods including the concentration of che218 inoculum solution was mentioned in Section 2.3. of  the revised manuscript.

                (See Section 2.3. in the revised manuscript, please)

L 112: What time of day was the treatment performed? What were the weather conditions before and after the treatment?

Answer: We treated che218 at 7 am on August 23, 2022 and August 4, 2023, respectively. The weather was fine before and after the treatment. These information was added in the revised manuscript.

                (See p. 3, line 127 in the revised manuscript, please)

L 115: part of the text is missing - please check it again.

Answer: Thank you for your suggestion. The part of the text was missing.

               We revised the text as follows:

               ‘Three bunches were collected for each treatment on days 0, 20, and 30 post-treatment in the 2022 growing season or days  0, 15, and 30 post-treatment in the 2023 growing season.’

                (See p. 3, lines 129-131 in the revised manuscript, please)

L 124: Was there a specific berry position in the bunch for sampling?

Answer: We collected six berries from the top of the bunch, eight berries from the middle of the bunch, and six berries from the bottom of the bunch (total  20 berries) from each bunch in the experiment. We added the information in the revised manuscript.

                (See p. 3, lines 140-141 in the revised manuscript, please)

L 133: Consider adding a formula instead of just mentioning it.

Answer: We appreciate your suggestion to include the formula instead of just mentioning it. However, due to the complexity of the formula, we believe it would be preferable not to include it in the revised manuscript, as it might potentially confuse readers.

It is important to add the climatic conditions during the seasons in which the field experiment was conducted.

Answer: Average temperatures, maximum and minimum temperatures, GDD, and precipitation from April 1 through October 31 in the experimental field in 2022 and 2023 growing seasons were added as supplemental Table S1 in the revised manuscript. We mentioned how the temperature differences between the two years affected anthocyanin accumulation promoted by che218.

               (See p. 2, lines 87-92 and p. 6, lines 243-257  in the revised manuscript and supplementary Table S1, please)

L 190: Consider adding the data in the supplementary material

Answer: We appreciate your suggestion to include the preliminary screening data in the supplementary material. However, as this was a preliminary screening experiment, the available images may not be of sufficient quality for readers to interpret clearly. Therefore, we prefer not to include this data in the revised manuscript. We hope you understand our decision.

L 213: Same comment as L 104

Answer: Our response to L104 addresses this comment.

                (See our response to L 104 in this response letter, please)

L 214: No need to add a new paragraph

Answer: As you suggested, we removed the new paragraph.

                (See p. 6, line 231 in the revised manuscript, please)

L 320: How about the weather conditions this season? Was it different from the previous season?

Answer: Average temperatures, maximum and minimum temperatures, GDD, and precipitation from April 1 through October 31 in the experimental field in 2022 and 2023 growing seasons were added as supplemental Table S1 in the revised manuscript. We mentioned how the temperature differences between the two years affected anthocyanin accumulation promoted by che218.

               (See p. 2, lines 87-92 and p. 6, lines 243-257  in the revised manuscript and supplementary Table S1, please)

L 361: to the best of our knowledge,

Answer: We changed ‘to our knowledge’ to ‘to the best of our knowledge’ in the revised manuscript.

                (See p. 11, lines 397-398 in the revised manuscript, please)

Add a final sentence highlighting the importance of the findings over these years of this beautiful research...

Answer: Thank you for your valuable suggestion. We have added another sentence in the revised Discussion section to highlight the importance of our findings. The added sentence emphasizes the significant contributions and long-term impact of our study in the field. We believe this addition strengthens the conclusion and provides readers with a clear understanding of the research's broader implications.

                (See the revised Discussion, please)

Reviewer 2 Report

Comments and Suggestions for Authors

Revision of manuscript microorganisms-3123506 “Anthocyanin accumulation in grape berry skin promoted by endophytic Microbacterium sp. che218 isolated from wine grape shoot xylem”

The manuscript is welcome, only few observations was made for improve the manuscript.

Section 2.3 Field experiments was writing in italics, is that correct?

Line 218, the authors mentioned with respect to the 2022 growing season… The effects of che218 on anthocyanin accumulation in berry skins became negligible on day 30 post-treatment… According to their figure 1B, the SCD control was superior than the che218, although not significantly but its evident. Why did the authors not considered the 30 day for mybA1 and UFGT transcription analysis in 2022 growing season?

The level in anthocyanin on day 30 in SCD control to what is attributable? Was there a great difference in weather in 2022 compared to season 2023. This can be discussed? This is because the level of anthocyanin in both seasons are comparable on 30 day with the che218 but not with the SCD control.

Author Response

Reviewer #2

Comments and Suggestions for Authors

Revision of manuscript microorganisms-3123506 “Anthocyanin accumulation in grape berry skin promoted by endophytic Microbacterium sp. che218 isolated from wine grape shoot xylem”

The manuscript is welcome, only few observations was made for improve the manuscript.

Answer: Thank you very much for your review to improve our manuscript. The following is our point-by-point response to the comments and detailing all changes made on the revised manuscript.

Section 2.3 Field experiments was writing in italics, is that correct?

Answer: Thank you for your feedback. The italics were a mistake, so we have corrected them.

                (See the revised Section 2.3, please)

Line 218, the authors mentioned with respect to the 2022 growing season… The effects of che218 on anthocyanin accumulation in berry skins became negligible on day 30 post-treatment… According to their figure 1B, the SCD control was superior than the che218, although not significantly but its evident. Why did the authors not considered the 30 day for mybA1 and UFGT transcription analysis in 2022 growing season?

Answer: As shown in Figure 2B, the difference between SCD and che218 on day 30 post-treatment was within the margin of error and was not statistically different, so we cannot agree that SCD is superior to che218 on day 30 post-treatment.  

Since anthocyanin biosynthesis is regulated by the expression of mybA1, which encodes a transcription factor that controls the expression of UFGT, a gene that encodes an enzyme catalyzing the glycosylation of anthocyanidins [Kobayashi  et al. 2004], we analyzed transcription levels of mybA1 and UFGT on day 20 post-treatment that the amount of anthocyanin accumulated in berry skins was different between SCD and che218 to determine whether che218 upregulated the transcription of mybA1 and UFGT.

The level in anthocyanin on day 30 in SCD control to what is attributable? Was there a great difference in weather in 2022 compared to season 2023. This can be discussed? This is because the level of anthocyanin in both seasons are comparable on 30 day with the che218 but not with the SCD control.

Answer: We appreciate your insightful question regarding the anthocyanin levels on day 30 in the SCD control and the potential influence of weather differences between the 2022 and 2023 seasons. To address this point, we have added supplementary Table S1 in the revised manuscript. This table provides detailed weather data for the experimental field, including average temperatures, maximum and minimum temperatures, Growing Degree Days (GDD), and precipitation from April 1 through October 31 for both the 2022 and 2023 growing seasons. In the revised manuscript, we have included a discussion on how the temperature differences between the two years affected anthocyanin accumulation, particularly in relation to the effects promoted by che218. This analysis helps to contextualize the observed differences in anthocyanin levels between the SCD control and che218-treated samples across both seasons.

                (See p. 2, lines 87-92 and p. 6, lines 243-257  in the revised manuscript and supplementary Table S1, please)

Reviewer 3 Report

Comments and Suggestions for Authors

The manuscript (MS) offers a valuable contribution by proposing an endophyte-based method to improve grape quality. The MS is generally good, however the following points need to addressed before acceptance:

1- The novelty and motivation of the wok need to be clearly highlighted.

2- Scientific names of plants should be used in the title of the MS.

3- The reason for not determining the species of Microbacterium should be  addressed.

4-  Why 2.3. Field experiments section is in italic?

5- References for Myb-related transcription factor 153 (MybA1) primers should be cited appropriately.

6- In Figure 5 all scientific names should be italic.

7-  Genome data of the Microbacterium sp should be presented in a much more way  with emphasis on Anthocyanin accumulation

8- the significance/conclusion of this study should be more elaborated.

  Comments on the Quality of English Language

Minor editing of English language required

Author Response

Reviewer #3

Comments and Suggestions for Authors

The manuscript (MS) offers a valuable contribution by proposing an endophyte-based method to improve grape quality. The MS is generally good, however the following points need to addressed before acceptance:

Answer: Thank you very much for your review to improve our manuscript. The following is our point-by-point response to the comments and detailing all changes made on the revised manuscript.

1- The novelty and motivation of the wok need to be clearly highlighted.

Answer: Thank you for your valuable suggestion. We have revised original final sentence to highlight the importance of our findings. The added sentence emphasizes the significant contributions and long-term impact of our study in the field. We believe this addition strengthens the conclusion and provides readers with a clear understanding of the research's broader implications.

                (See p. 11, lines 416-433 in the revised manuscript, please)

2- Scientific names of plants should be used in the title of the MS.

Answer: As mentioned in the manuscript, che218 was indeed isolated in our previous study [Hamaoka et al. 2021]. Adding the scientific name of wine grape, Vitis vinifera, to the paper's title would make it excessively long. Therefore, we prefer to retain the current title.

3- The reason for not determining the species of Microbacterium should be  addressed.

Answer: We would like to draw your attention to the fact that this information is already addressed in our original manuscript. Specifically, in Section 3.4., we have provided a detailed explanation of why we were unable to definitively determine the species of che218. We believe this explanation in the original manuscript sufficiently addresses your  concern.

                (See p. 8, lines 318-328 in the revised manuscript, please)

4-  Why 2.3. Field experiments section is in italic?

Answer: Thank you for your feedback. The italics were a mistake, so we have corrected them.

                (See the revised Section 2.3., please)

5- References for Myb-related transcription factor 153 (MybA1) primers should be cited appropriately.

Answer: Thank you for your suggestion. We designed primers for qRT-PCR from the CDS region of MybA1 in the sequence of 'Vitis vinifera gypsy-type retrotransposon Gret1 DNA and VvmybA1 gene for myb-related transcription factor, complete cds, cultivar: Ruby Okuyama' (GenBank accession no. AB111101). Therefore, the current citation (GenBank number) is correct.               

                (See p. 4, lines 170-172 in the revised manuscript, please)

6- In Figure 5 all scientific names should be italic.

Answer: As you pointed out, the scientific names of bacteria tested are italic in the revised Figure 5.

                (See the revised Figure 5, please)

7-  Genome data of the Microbacterium sp should be presented in a much more way  with emphasis on Anthocyanin accumulation

Answer: We sincerely appreciate the reviewer's insightful comments. We have addressed that che218 does not possess genes related to abscisic acid synthesis, which are involved in anthocyanin production in the revised manuscript. As suggested, we have expanded our discussion on the genome data of che218 with a particular emphasis on genes potentially related to anthocyanin accumulation. On the other hand, we acknowledge that the presence of similar genes in other Microbacterium sp. genomes does not necessarily indicate their ability to promote anthocyanin synthesis. In future research directions, we will plan a) Elucidating the specific mechanisms by which che218 promotes anthocyanin synthesis and  b) Conducting comparative genomic analyses of che218 and other Microbacterium sp. to identify potential genetic factors involved in anthocyanin accumulation.

8- the significance/conclusion of this study should be more elaborated.

Answer: Thank you for your valuable suggestion. We have added another sentence in the revised Discussion section to highlight the importance of our findings. The added sentence emphasizes the significant contributions and long-term impact of our study in the field. We believe this addition strengthens the conclusion and provides readers with a clear understanding of the research's broader implications.

                (See the revised Discussion, please)

Reviewer 4 Report

Comments and Suggestions for Authors

Article: Anthocyanin accumulation in grape berry skin promoted by endophytic Microbacterium sp. che218 isolated from wine grape shoot xylem.

This work is devoted to the study of the effect of grape endophytic bacteria on the ability of grapes to accumulate anthocyanins. The study of the effect of endophytes on plants is an actively developing area, the results of which are sometimes extremely interesting and promising. The design of the experiment is adequate, but in my opinion it could be improved (specific suggestions are given below). The materials and methods are described well, but there is a little confusion with the description of the anthocyanin extraction method. All figures are presented in good quality, and statistical data processing is beyond doubt. The manuscript states that there are additional materials, but I could not evaluate them due to their absence in the system. Perhaps the authors forgot to upload them or this is a system error. Below I provide my suggestions and comments in detail. Also, in the text I recommend using numbers for mass, volume and quantity. Overall, despite the shortcomings, this work is interesting and certainly deserves publication.

Introduction. Overall, everything is good, but we can discuss the approach to increasing the anthocyanin content through the RNA interference mechanism, using exogenous dsRNA.

Also, try to describe the putative mechanism through which the phenylpropanoid pathway is affected.

Materials and methods. Specify the type and variety of grapes from which the endophytic bacterium was isolated (Line 88).

Why is Chapter 2.3. Field experiments entirely in italics?

Specify the weather conditions during the experiments in 2022 and 2023 (Line 112).

2.4. Anthocyanin measurement. Typically, 1% HCl in methanol at 4°C is used to extract anthocyanins. Are you sure you used 1% methanol? Then you talk about 1% HCl, but do not mention methanol. Put the description in order.

2.7. Real-time RT-PCR. Why was only one housekeeping gene β-actin used to normalize the expression data? For reliable normalization, it is recommended to use several genes, which has been repeatedly confirmed by studies.

2.9. Statistical analysis. How many biological and technical replicates were there?

3. Results. 3.3. Effect of che218 treatment on the transcription of anthocyanin biosynthesis-related genes in 248 berry skins of field-grown grapevines. Why was the analysis limited to the MybA1 and UFGT genes? It would be interesting to look at the expression of early anthocyanin biosynthetic genes (4CL, CHS, CHI, F3H) and later DFR and ANS. Analysis of these genes may allow a more complete and thorough assessment of the effects of the che218 bacterium on anthocyanin accumulation and secondary metabolism in general. For example, it would be worth assessing the level of accumulation of resveratrol and its derivatives in grapes, since this is a very popular and interesting group of substances in grapes.

Discussion. Overall, well written, but a broader range of results would have made the discussion more interesting.

Author Response

Reviewer #4

Comments and Suggestions for Authors

This work is devoted to the study of the effect of grape endophytic bacteria on the ability of grapes to accumulate anthocyanins. The study of the effect of endophytes on plants is an actively developing area, the results of which are sometimes extremely interesting and promising. The design of the experiment is adequate, but in my opinion it could be improved (specific suggestions are given below). The materials and methods are described well, but there is a little confusion with the description of the anthocyanin extraction method. All figures are presented in good quality, and statistical data processing is beyond doubt. The manuscript states that there are additional materials, but I could not evaluate them due to their absence in the system. Perhaps the authors forgot to upload them or this is a system error. Below I provide my suggestions and comments in detail. Also, in the text I recommend using numbers for mass, volume and quantity. Overall, despite the shortcomings, this work is interesting and certainly deserves publication.

Answer: Thank you very much for your review to improve our manuscript. The following is our point-by-point response to the comments and detailing all changes made on the revised manuscript.

Introduction. Overall, everything is good, but we can discuss the approach to increasing the anthocyanin content through the RNA interference mechanism, using exogenous dsRNA.

Answer: Thank you for your insightful comment regarding the potential use of RNA interference mechanisms to increase anthocyanin content in plants. We appreciate the suggestion to discuss this approach using exogenous dsRNA. While the application of exogenous double-stranded RNAs (dsRNAs) or small-interfering RNAs (siRNAs) has shown promise in various plant species (Arabidopsis, tomato etc) for modulating gene expression and metabolic pathways, we must note that, to the best of our knowledge, there are currently no conclusive reports demonstrating the efficacy of these methods specifically for promoting anthocyanin accumulation in grape berry skins. The use of RNA interference techniques in viticulture, particularly for enhancing anthocyanin content, represents an intriguing area for future research. We thank the reviewer for bringing this perspective to our attention.

Also, try to describe the putative mechanism through which the phenylpropanoid pathway is affected.

Answer: As you suggested, we have included information regarding the effect of exogenous ABA application on grape berry skins. Specifically, we noted that such application increased the expression of flavonoid synthesis genes in the phenylpropanoid pathway, citing the work of Koyama et al. (2018) [reference number 11] in the revised Introduction section.

                (See p. 2, lines 50-51 in the revised manuscript, please)

Materials and methods. Specify the type and variety of grapes from which the endophytic bacterium was isolated (Line 88).

Answer: We added the cultivars of grapevines from which the endophytic bacterium was isolated in the revised Materials and methods section.

                (See p. 3, lines 96-97 in the revised manuscript, please)

In addition, we mentioned that che218 was isolated from Chardonnay in the revised Results section.

                (See p. 5, line 208 in the revised manuscript, please)

Why is Chapter 2.3. Field experiments entirely in italics?

Answer: Thank you for your feedback. The italics were a mistake, so we have corrected them.

                (See the revised Section 2.3., please)

Specify the weather conditions during the experiments in 2022 and 2023 (Line 112).

Answer: Average temperatures, maximum and minimum temperatures, GDD, and precipitation from April 1 through October 31 in the experimental field in 2022 and 2023 growing seasons were added as supplemental Table S1 in the revised manuscript. We mentioned how the temperature differences between the two years affected anthocyanin accumulation promoted by che218.

                (See p. 2, lines 87-92 and p. 6, lines 243-257  in the revised manuscript and supplementary Table S1, please)

2.4. Anthocyanin measurement. Typically, 1% HCl in methanol at 4°C is used to extract anthocyanins. Are you sure you used 1% methanol? Then you talk about 1% HCl, but do not mention methanol. Put the description in order.

Answer: Those were our mistake. We used ‘1% HCl–methanol’ as an extraction solvent.  We revised the Section  2.4. in the revised manuscript.

                (See the revised Section 2.4., please)

2.7. Real-time RT-PCR. Why was only one housekeeping gene β-actin used to normalize the expression data? For reliable normalization, it is recommended to use several genes, which has been repeatedly confirmed by studies.

Answer: We used 18S rRNA and β-actin as internal controls (for example, Kobayashi et al. 2009. American Journal of Enology and Viticulture 60:362-367.). Since β-actin was recommended as a reference gene during grape berry development (Reid et al., 2006), β-actin was used for normalization as a reference gene and then expression levels of each gene were expressed as relative values to β-Actin gene expression level in this study.

                   *Reid, K. E., Olsson, N., Schlosser, J., Peng, F., & Lund, S. T. (2006). An optimized grapevine  RNA isolation procedure and statistical determination of reference genes for real-time RT-PCR during berry development. BMC plant biology, 6(1), 27.

2.9. Statistical analysis. How many biological and technical replicates were there?

Answer: We would like to explain that the number of biological replicates varied across our different experimental systems. To address this variation and provide clear, specific information to readers, we have not included a general statement about the number of biological replicates in Section 2.9 (Statistical analysis) of the Materials and Methods. Instead, we have provided detailed information on the number of biological replicates for each experiment in the corresponding figure legends.              

 (See each Figure legend in the revised manuscript, please)

  1. Results. 3.3. Effect of che218 treatment on the transcription of anthocyanin biosynthesis-related genes in 248 berry skins of field-grown grapevines. Why was the analysis limited to the MybA1 and UFGT genes? It would be interesting to look at the expression of early anthocyanin biosynthetic genes (4CL, CHS, CHI, F3H) and later DFR and ANS. Analysis of these genes may allow a more complete and thorough assessment of the effects of the che218 bacterium on anthocyanin accumulation and secondary metabolism in general. For example, it would be worth assessing the level of accumulation of resveratrol and its derivatives in grapes, since this is a very popular and interesting group of substances in grapes.

Answer: We appreciate your insightful comment regarding the scope of our gene expression analysis and the suggestion to explore other phenylpropanoid pathway genes. Our focus on MybA1 and UFGT genes was deliberate, as these genes play crucial roles in determining anthocyanin accumulation in grape berry skin. MybA1 is a key transcription factor that regulates the expression of anthocyanin biosynthesis genes, while UFGT catalyzes the final step in anthocyanin biosynthesis, which is often considered the rate-limiting step in this pathway. The expression levels of these two genes are highly correlated with anthocyanin content in grape berry skins.

                     We agree that analyzing the expression of early anthocyanin biosynthetic genes (4CL, CHS, CHI, F3H) and later genes (DFR and ANS) would provide a more comprehensive view of che218 effects on grape berry qualities. Similarly, assessing the accumulation of resveratrol and its derivatives is an intriguing suggestion, given their importance in grape research. While our current study focused primarily on anthocyanin accumulation, we acknowledge the value of a broader investigation into secondary metabolism. In response to this comment, we have added a new paragraph in the Discussion section of our revised manuscript. This addition addresses the potential effects of che218 on other functional compounds such as resveratrol and flavonoids, and highlights this as an important area for future research.

                (See the revised Discussion, please)

Discussion. Overall, well written, but a broader range of results would have made the discussion more interesting.

Answer: Thank you for your valuable suggestion. We have added another sentence in the revised Discussion section to highlight the importance of our findings. The added sentence emphasizes the significant contributions and long-term impact of our study in the field. We believe this addition strengthens the conclusion and provides readers with a clear understanding of the research's broader implications.

                (See the revised Discussion, please)

Round 2

Reviewer 4 Report

Comments and Suggestions for Authors

Thanks to the authors for the kind and constructive response. In general, the authors have done everything they could without adding new data to the work. The work has become better.

Author Response

Thank you very much for taking the time to review our manuscript. Your insightful comments and suggestions have been invaluable in improving the quality of our work.